# Is Personal Protective Equipment Worth the Hassle? Annual Risk of Cryptosporidiosis to Dairy Farmers and How Personal Protective Equipment and Handwashing Can Mitigate It

**DOI:** 10.3390/microorganisms11102413

**Published:** 2023-09-27

**Authors:** Alexis L. Mraz, Nandini Mutyala, Seana Cleary, Brenda F. Seals

**Affiliations:** 1The Department of Public Health, School of Nursing and Health Sciences, The College of New Jersey, Ewing Township, NJ 08618, USA; nm3403@cumc.columbia.edu (N.M.); sealsb@tcnj.edu (B.F.S.); 2The Department of Epidemiology, Mailman School of Public Health, Columbia University, New York, NY 10032, USA; 3The Gangarosa Department of Environmental Health, Rollins School of Public Health, Emory University, Atlanta, GA 30322, USA

**Keywords:** quantitative microbial risk assessment (QMRA), dairy farms, cryptosporidiosis, personal protective equipment (PPE), zoonotic diseases, *Cryptosporidium parvum*

## Abstract

Cows are known carriers of *Cryptosporidium parvum* (*C. parvum*), a protozoa that can cause the gastrointestinal illness cryptosporidiosis in humans. Despite this potential exposure, dairy farmers tend to wear personal protective equipment (PPE) to protect the milk from contamination, rather than to protect themselves from zoonotic diseases, such as cryptosporidiosis. In this study, cow feces were collected from individual cattle on dairy farms and analyzed for *C. parvum* using qPCR. Quantitative microbial risk assessment (QMRA) was used to determine the risk of cryptosporidiosis to the dairy farmer with and without the use of handwashing and PPE (gloves and masks). The annualized risk of cryptosporidiosis to dairy farmers was 29.08% but was reduced significantly in each of the three interventions. Among the individual interventions, glove use provided the greatest reduction in risk, bringing the annual risk of cryptosporidiosis to 4.82%. Implementing regular handwashing, the use of gloves and a mask brought the annual risk of cryptosporidiosis to 1.29%. This study provides evidence that handwashing and PPE use can significantly reduce the risk of cryptosporidiosis to farmers and is worth implementing despite potential barriers such as discomfort and cost.

## 1. Introduction

*Cryptosporidium parvum* (*C. parvum*) and *Cryptosporidium hominis* (*C. hominis*) are protozoa that cause cryptosporidiosis in humans, a parasitic infection resulting in gastrointestinal illness, mainly diarrhea, stomach cramps, dehydration, nausea, vomiting, fever, and weight loss [1]. Typically, cryptosporidiosis is self-limiting in healthy individuals, but may be chronic or fatal in immunocompromised people [1]. The case fatality risks are roughly 29% in the immunodeficient, with a range of 19–46% in AIDS patients [2,3]. *Cryptosporidium* is the leading cause of waterborne disease outbreaks and the third leading cause of zoonotic enteric illness in the US, with an estimated 748,000 cases of cryptosporidiosis annually, resulting in over USD 45 million in hospitalization costs [4,5,6,7]. An estimated 15–50% of the adult US population is seropositive for *Cryptosporidium* [8,9,10]. New Jersey had 1.8 identified cases per 100,000 residents in 2018 [11]. 

As a protozoa, *Cryptosporidium* is able to survive for long periods in the environment and is resistant to desiccation and oxidative stress [1]. It can be transmitted through contact with soil, food, water, or surfaces that have been contaminated with the feces of infected humans or animals. Cryptosporidiosis was first reported among cattle in the early 1970s and is now recognized as an endemic illness in cattle on a global scale [2]. Four species of *Cryptosporidium* are commonly found in cows, but only *C. parvum* is associated with clinical disease in both humans and neonatal calves and oocyst shedding in older cows [2]. *C. parvum* is very contagious and can be common among dairy cattle; in some cases it has been found in the feces of 70% of dairy calves [3]. *C. parvum* causes neonatal enteritis in calves, producing symptoms such as profuse watery diarrhea, lack of appetite, lethargy, and dehydration. Without proper treatment, illness can lead to death for this population. Calves shed large numbers of oocysts 4 to 12 days after infection, which are highly infectious and can lead to sickness in other susceptible cows [2]. Infection typically occurs within the first week of life and its high infectivity can make it difficult to manage on a farm. In the U.S., treatment is palliative by providing fluid and electrolyte replacement as well as nutritional support and antidiarrheal remedies [3]. Zoonotic transmission of *C. parvum* may occur if a person comes into direct or indirect contact with infected cows or cow feces, which can then lead to person-to-person transmission, increasing the potential for an outbreak. 

The majority of human cases of cryptosporidiosis are caused by *C. parvum*, the zoonotic species, or *C. hominis*, the human-adapted species [2]. While there is global prevalence of the disease, it is most often detected in developing countries with poor sanitary conditions due to the fecal–oral pathway of transmission of the parasite. In the United States, cryptosporidiosis is a nationally notifiable disease and from 2009 to 2017, there were 444 *Cryptosporidium* outbreaks, which resulted in over 7000 cases [5]. Contact with cattle was associated with 65 of the 444 outbreaks, which led to 549 cases of cryptosporidiosis. However, the Centers for Disease Control and Prevention estimates there to be 823,000 cases of cryptosporidiosis on an annual basis [6]. This significant underreporting of the disease is most likely due to lack of diagnostic testing and minimal reporting of cases to the health authorities. Animal handlers are at an increased risk of contracting *C. parvum* due to their frequent contact with cattle and the potential of transmission from infected calves, which warrants this study on the risk analysis of *C. parvum* and the health of dairy farmers.

Dairy farming plays a significant role in our society, producing billions of pounds of milk per year and contributing to the economy as a major commodity and as a source of income for thousands of people in the U.S. As of 2020, there are about 32,000 licensed dairy operations [7], with 94% of these farms being family owned and operated [8]. Farms are often passed down within a family for several generations. According to U.S. Department of Agriculture’s data from 2017, about 31% of dairy farms have between 1 and 9 cows, about 21% have between 10 and 49 cows, and 22% have between 50 and 99 cows. Only 3.5% of dairy operations oversee herd sizes of greater than 999 cows [9]. In this study, New Jersey (NJ) dairy farms were observed for PPE use and fecal samples were taken from the cows to test for the presence of *C. parvum*. Across the country, New Jersey ranks 45th in total state milk production and produces approximately 90 million pounds of milk per year [10]. The state is home to about 35 dairy farms with an average herd size of 114 cows [10]. Considering the sizable number of individuals involved in the dairy industry who frequently interact with cattle, it is essential to examine the risks associated with the practice of dairy farming and the measures that can be taken to mitigate these risks.

The Occupational Safety and Health Administration (OSHA) established several regulations for the dairy industry to ensure safe and healthy working conditions for dairy farm workers. These standards include a list of PPE to prevent injuries to the workforce. Eye protection, such as goggles or safety glasses, are utilized to protect the eyes from cows swatting their tails while being milked. Gloves and aprons prevent exposure to chemicals that are used during the pre-milking process and when there is the potential of workers contacting animal body fluid. Work boots with toe caps help protect the feet from being stepped on by cow hooves [11]. Not only does the use of PPE minimize the risk of exposure to infectious pathogens for dairy farm workers, but it also reduces the likelihood of milk quality issues and cross-contamination. Prior studies have shown that the presence of bacteria on used gloves is 75% less than on bare hands and wearing gloves reduces the spread of infectious and environmental bacteria by 50% [12]. While most dairy farmers are often knowledgeable regarding the protective factors of PPE, there is a discrepancy in the utilization of such measures in the workplace.

Also note that as the demands for the dairy industry increase, operations are frequently relying upon the immigrant workforce to meet worker shortages and maintain adequate production and supply. However, due to the limitations of reporting systems, negative perceptions of immigration agencies, and fear of employer consequences, minimal data are available on incidence and prevalence of workplace injuries and illnesses among immigrant dairy farm workers [13]. Menger et al.’s research on the perceptions of health and safety among immigrant Latinx dairy workers found that PPE availability varied by facility. Immigrant workers were resistant to using PPE due a lack of full understanding of the risks on the farm, negative impacts on job performance, and perceived inconveniences [13]. Since these safety measures were not implemented in their country of origin, workers often felt uncomfortable with the use of PPE while at work. Another study on the views of health and safety among immigrant dairy farm workers discovered that PPE was only provided after there had been a harmful exposure or injury [14]. There were still instances where, even after hazards were identified, the employer did not immediately resolve the issue or provide the necessary tools to protect the safety and wellbeing of their employees. Importantly, the issues surrounding PPE use revolve around both the dairy farm workers and upper management directing workers. Without the proper protocols, management/worker issues increase the likelihood of zoonotic transmission of infectious agents, which can potentially spread to a large population and lead to severe negative consequences. The goal of this study was to assess the risk of cryptosporidiosis among dairy farmers and determine the extent to which that risk is mitigated when farmers wear various PPE and/or engage in regular handwashing practices.

## 2. Materials and Methods

### 2.1. Dairy Farm Selection and Sampling

New Jersey has 35 active dairy farms, all of which were contacted to request participation in this study. Eight agreed to participate. One dairy farm dropped out of this study before samples were collected due to a worker’s illness. Seven dairy farms were sampled with herds ranging from 8–130 cattle. A total of 70 samples were collected, ranging from 5–22 samples per farm. Samples were collected from every animal observed defecating during the time of sampling. Sampling time ranged from 2–4 h at each farm depending on farmer availability. Samples were collected from individual cattle directly after the animal was observed to defecate. Samples were marked with the animal identification, age, and any information about diarrhea or other gastrointestinal illnesses. Only one sample was obtained per individual animal. Samples were transported to the laboratory at The College of New Jersey on ice and stored at −20 °C within 6 h [15,16]. Sampling took place from June–September of 2021.

### 2.2. Laboratory Analysis

DNA extraction was performed using a Zymo Research Quick-DNA^TM^ Fecal/Soil Microbe Miniprep Kit, Irvine, CA, USA as per kit instructions using 0.2 g of feces per sample. DNA extracts were analyzed for *C. parvum* using quantitative polymerase chain reaction (qPCR) within 30 h of extraction. The direct and reverse primer sequences were CATGGATAACCGTGGTAAT and TACCCTACCGTCTAAAGCTG, respectively, with a *C. parvum* specific probe 56-FAM/ATCACATTAAATGT/3MGBEc/ with a primer/probe concentration of 300 nM/50 nM [17,18]. The qPCR consisted of a 95 °C, 5 min hot start followed by 45 cycles of 95 °C for 10 s and 60 °C for 60 s [18]. Samples were run in triplicate.

### 2.3. QMRA

QMRA, a quantitative approach to assessing public health risk by estimating risk of infection and illness when a population is exposed to a pathogen in the environment, is conducted in four steps. These include (a) hazard identification, (b) exposure assessment, (c) dose response, and (d) risk characterization [19]. This QMRA assesses the risk of cryptosporidiosis to dairy farmers from the ingestion of *C. parvum* through a fecal–oral pathway. The amount of *C. parvum* ingested is modeled assuming the transfer of cow feces from the farmer’s hand to mouth using the parameters in Table 1. The limit of quantification was determined from the standard curve of the qPCR assay used to measure the concentration of *C. parvum* in the cow feces in this study. Samples that were positive for *C. parvum* but below the limit of quantification were assessed at half the limit of quantification. Samples in which *C. parvum* was not detected during qPCR were assessed at half the limit of detection. The limit of detection was defined as 3 gene copies (gc) per qPCR reaction [20]. Evers et al. (2014) modeled the amount of feces transferred to an adult’s lips while on a farm using pre-applied cow feces and *E. coli* WG5 as an indicator, which was used in this model for the ingestion value [21]. The concentration of *C. parvum* oocyst per gram of feces (*Cp_Conc_*) was calculated using the geometric mean of the gene copy per gram of feces sample concentrations (*Cp_gc_*) in this study divided by the average number of gene copies per *C. parvum* oocyst (*Cp_gc/oocyst_*) (Equation (1)) [22].
(1)$${Cp}_{Conc}={Cp}_{gc}/{Cp}_{gc/oocyst}$$

**Table 1 microorganisms-11-02413-t001:** Parameters used in the quantitative microbial risk assessment (QMRA) model.

Variable	Value	Unit	Distribution	Reference
Limit of quantification	1.53 × 10^3^	gc/gram	Point estimate	This study
Limit of detection	2.50 × 10^2^	gc/gram	Point estimate	This study [20]
gc/oocyst	13.5	-	Point estimate	[22]
Farmer’s daily ingestion of feces	Mean: 6.97 × 10^−5^ Std: 5.74 × 10^−4^	grams	Normal distribution	[21]
*C. parvum* in cow feces	Geometric mean: 2.31 × 10^3^ Geometric Std: 2.66	oocysts/ gram	Normal distribution	This study
k	5.72 × 10^−1^	Parameter	Exponential dose response	[23]
Probability of illness given infection	Min: 2.00 × 10^−1^ Max: 7.00 × 10^−1^	-	Uniform distribution	[22]
*C. parvum* reduction from handwashing	1.00 × 10^−2^	-	Point estimate	[24]
Reduction in mouth touches when wearing a mask	Mean: 3.14 × 10^−1^ Std: 1.37 × 10^−2^	-	Normal distribution	[25]
Reduction in mouth touches when wearing gloves	Mean: 2.01 × 10^−1^ Std: 1.28 × 10^−1^	-	Normal distribution	[26]
Transfer of pathogens to hands when doffing gloves	Min: 3.00 × 10^−7^ Max: 1.00 × 10^−1^	-	Uniform distribution	[27]

The daily dose of *C. parvum* (*D_Cp_*) the farmer ingests in oocysts was calculated by multiplying the concentration of *C. parvum* oocyst per gram of feces by the amount of feces ingested (*F_I_*) in grams (Equation (2)).
(2)$$D_{Cp}={Cp}_{Conc}*F_{I}$$

An exponential dose response with an endpoint of infection (*P_inf_*) was calculated where *k* is the probability of the pathogen surviving to initiate infection in the host and *Dcp* is the dose ingested (Equation (3)) [23]. The probability of illness was calculated by multiplying the probability of infection by the probability of illness given infection, modeled from a uniform distribution (Equation (4)) [22].
(3)$$P_{inf}=1-e^{-k*Dcp}$$
(4)$$P_{ill}=P_{inf}*P_{ill/inf}$$

Risks were annualized (*AR*) with an assumption of 5 farming days per week for 50 weeks per year (*FD_a_*) (Equation (5)).
(5)$$AR=1-(1-{P_{ill})}^{FDa}$$

### 2.4. Risk Reduction from PPE Use

The mitigation of cryptosporidiosis risk was calculated for farmers when using gloves, face masks, and handwashing by calculating the reduction in pathogen transfer to the lips. The dose of *C. parvum* the farmer ingests while wearing gloves (*D_G_*) was calculated by multiplying the daily dose of *C. parvum* (*D_Cp_*) by the reduction in mouth/lip touches when a farmer is wearing gloves (*T_G_*), or the proportion of pathogens transferred to the hand while doffing the glove (*H_tran_*) (Equation (6)). A binomial distribution was used to decide if a farmer was wearing gloves or not at the point of the face touch.
(6)$$D_{G}=\left(D_{Cp}*T_{G}\right)\cap \;\left(D_{Cp}*H_{tran}\right)$$

The dose of *C. parvum* the farmer ingests while wearing a mask (*D_M_*) was calculated by multiplying the daily dose of *C. parvum* (*D_Cp_*) by the reduction in mouth/lip touches when a farmer is wearing a mask (*T_M_*) (Equation (7)).
(7)$$D_{M}=\left(D_{Cp}*T_{M}\right)$$

The dose of *C. parvum* the farmer ingests when engaging in regular handwashing (*D_HW_*) was calculated by multiplying the daily dose of *C. parvum* (*D_Cp_*) by the *C. parvum* reduction during handwashing (*HW_red_*) (Equation (8)) [24]. A binomial equation was used to determine if the farmer had touched their face before or after handwashing.
(8)$$D_{HW}=\left(D_{Cp}*{HW}_{red}\right)\cap \;D_{Cp}$$

To calculate the reduction in risk when combining handwashing and glove use, the glove use model was first used to determine reduction in face touches and transfer of pathogens to hands while wearing gloves. A further reduction in *C. parvum* due to handwashing was then applied to the pathogens remaining on the hands after the gloves were doffed. Touches to the face were further reduced when mask wearing was added to both glove wearing and handwashing.

## 3. Results

### 3.1. C. Parvum Prevalence and Concentration on Farms

*C. parvum* was detected in 12.9% (9/70) of sampled cow feces with an average of 1.29 × 10^5^ and a median of 8.82 × 10^3^ oocysts/gram of feces. Adults comprised 65.2% (45/70) of the cattle samples and calves (<1 year of age) comprised the remaining 35.7% (25/70). Young calves under 1 month made up 14.3% (10/70) of the samples. Only two sampled cows (2/70, 2.9%) showed symptoms of cryptosporidiosis through diarrhea, including one 4-month-old and one 2-week-old calf. The older calf’s feces were not positive for *C. parvum*, while the younger calf had the highest concentration of *C. parvum* in this study at 1.06 × 10^6^ oocysts/gram. Of the samples that tested positive for *C. parvum*, 55.6% (5/9) were adult cows, 22.2% (2/9) were calves older than one month, and 22.2% (2/9) were young calves younger than one month. Positive samples were found at 57.1% of the farms (4/7) with each farm having only 1–2 cattle positive for cryptosporidiosis.

### 3.2. Risk of Cryptosporidiosis to Dairy Farmers and Mitigation from PPE

The annual risk of becoming ill with cryptosporidiosis for dairy farmers that do not use PPE or engage in regular handwashing is 29.08% [95% CI: 28.96–29.20]. This risk is reduced to 11.09% [95% CI: 11.03–11.14] when farmers wear a mask, 4.82% [95% CI: 4.78–4.86] when they wear gloves, and 14.72% [95% CI: 14.60–14.84] when they wash their hands regularly. A combination of wearing gloves and washing hands after doffing the gloves further reduces risk to 3.88% [95% CI: 3.83–3.92] and engaging in all three interventions results in the lowest risk of 1.29% [95% CI: 1.28–1.30]. The annual risks are displayed using boxplots in Figure 1 and ridge plots in Figure 2 to show the change in distribution of the annual risk among interventions.

## 4. Discussion

### 4.1. Farmers’ Attitude towards PPE and Zoonotic Disease

In the researchers’ observations at the farms sampled for this study, PPE use was infrequent and predominantly occurred to protect the milk during the milking process rather than to protect the farmer during activities such as “mucking out” barns where there was more likely to be contact with fecal manner. Discussions with the farmers mirrored attitudes seen in the literature. No farmer expressed much concern over zoonotic diseases or the potential for cryptosporidiosis. Most farmers were confident that none of their cattle would test positive for cryptosporidiosis. Most attributed their confidence to their cattle being fed well, having room to roam, and/or having clean stalls or barns. At one of the larger farms, the farmer discussed part of their herd contracting giardiasis and/or cryptosporidiosis the winter before we sampled. They had a veterinarian treat the sick animals, but some calves did die as a result. They were very vigilant for signs of gastrointestinal disease, such as diarrhea, to avoid another outbreak. Animals with symptoms were housed away from the herd and treated.

The farmers were more concerned with the risk of their cattle contracting cryptosporidiosis than contracting the disease themselves. It was hard to assess from conversation if that was due to a perception of cryptosporidiosis not being serious in humans or the perception that it would be difficult to contract from the cattle. Attitudes surrounding PPE and contact with feces were cavalier at most farms. Occasionally farmers referred to the researchers’ wearing gloves while collecting samples with phrases such as:

“When you do this long enough, you get used to it; it’s such a little poop.”

“I can grab it [fecal sample] for you if you’re uncomfortable.”

“You do this every day, and it becomes more of a hassle than it’s worth” in reference to a researcher breaking a glove while putting it on.

Sampling took place in summer 2021 while recommendations from the COVID-19 pandemic still suggested wearing face masks indoors. Most farmers did not reference the researchers wearing face masks. The one farmer that did, did so in the context of the pandemic. No farm workers were observed wearing face masks during this study.

Handwashing took place at varying degrees on each farm. Some farms had readily available handwashing stations in multiple locations while others were not as convenient or evident. Famers were inconsistent with washing their hands after interaction with cattle but were very consistent in washing their hands before going to the barn, particularly before milking.

PPE to prevent physical injury was common. Closed-toed shoes were worn by every farm worker observed, and many wore steel-toed and anti-slip boots. Long hair was consistently tied back.

### 4.2. PPE, Why We Do Not Use It, and Why We Should

PPE is not always comfortable, especially during hot days. Hands sweat more in gloves, which can make glove replacement challenging when hands are sweaty. Masks can cause glasses to fog up and/or make a worker feel warmer or more uncomfortable in hot weather. One study found that among women on dairy farms in Pennsylvania, 37.2% of the sample (out of 3709 participants) claimed to wear gloves when milking and about 71% always washed their hands after milking. The majority of the respondents to the survey believed that they had minimal risk of contracting a disease from the animals they came into contact with, which explains why PPE use was infrequent [28]. Another study found that only 43% of dairy operations required workers to wear boots and 39% to use gloves when handling pre-weaned cows among the farms that were surveyed [29]. Additionally, PPE presents a perceived economic and environmental issue for many farmers who view PPE as something else to buy and throw out.

The inclination to not wear PPE is understandable, which is why quantifying risk mitigation from PPE is so important. This study shows that the combination of glove use, mask use, and handwashing is the most effective way to reduce the risk of cryptosporidiosis. The most substantial single-intervention reduction is glove use with an annual risk reduction of 24.3%. Adding the use of masks and regular handwashing further reduced the annual risk by 3.6%. While promoting the use of all three interventions is ideal, it is recommended that farm owners or managers encourage employees to wear gloves, wash their hands frequently, and limit touching their faces as much as possible in situations where that is not realistic.

### 4.3. Study Limitations

As a pilot study, the sample size was small, including 70 samples from 7 different farms. Recruiting farms for participation was challenging. Some farm owners were skeptical of the researchers’ true reason for sampling at the farm while others did not have the time or manpower to facilitate sampling. Participation bias was a factor in this study as well. All the sampled farms had fewer than 150 cattle and were family owned and operated. Larger corporate farms did not agree to participate. All participating farms allowed their cattle access to open pastures, and barns and stalls appeared clean. The results of this study may not be generalizable to much larger farms, farms in other regions, or farms with other grazing and/or pasture practices.

Researchers did not conduct formal farmer interviews or surveys to assess their views of zoonotic diseases or PPE. Instead, researchers opted for a more naturalistic environment hoping to put farmers at ease so normal practices could be observed. This limited available data to casual conversation; hence, inferences are suggestive, not conclusive. Given difficulties in recruitment, researchers were concerned that a formal survey could cause doubts among participants that would influence their PPE use and hand hygiene while researchers were at the farm. Having no formal survey did not allow for a systematic analysis of the qualitative data recorded. However, this pilot study is one of the first of its kind, laying ground for a standardized protocol that could be implemented once trust has been established.

## 5. Conclusions

Our observational study documented common farmer practices consistent with past research that was primarily based on self-reports. Findings suggest that farmers experience significant risk of exposure to zoonotic disease that can result in health consequences. The observed practices and lack of reliance on PPE suggest that these occupational settings create conditions that allow for outbreaks of disease such as that reported by one farmer in the study. Our study results affirm the importance of prevention practices for farmers, including the following:
PPE use is effective in reducing the risk of cryptosporidiosis to dairy farmers;Handwashing and PPE use should be utilized to protect farmers from zoonotic disease and thought of as a protection to both the farmer and the product;Handwashing should occur frequently throughout the day when farmers are working with cattle;The use of gloves and masks should be encouraged in addition to frequent handwashing.

Public health practitioners concerned with infectious disease should prioritize research that includes direct observation and sampling as in this study. Innovation in prevention measures should explore working with PPE companies to make PPE more comfortable and convenient to encourage ease of adoption in farmers. Health education regarding zoonotic disease should reframe messages to highlight the benefits of PPE and handwashing in protecting farmers’ health and the farmers’ role in preventing disease spread to vulnerable family members and friends, such as those who are immunocompromised or have underlying health conditions. Health education should also extend to immigrant farm workers using communication channels that would highlight benefits of using PPE in culturally competent ways. Only by making prevention part of everyday life will farms be afforded protection from animal and human disease outbreaks from potential vectors.

## Figures and Tables

**Figure 1 microorganisms-11-02413-f001:**
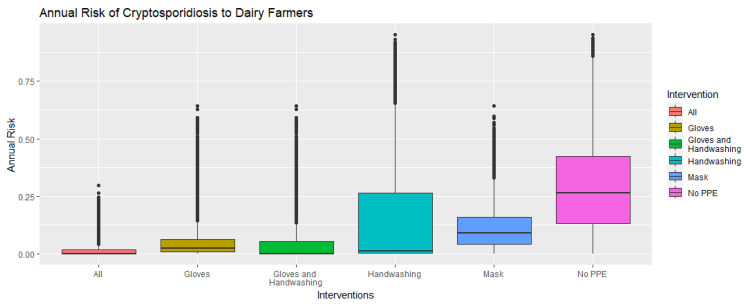
Boxplot displaying annual risk of cryptosporidiosis to dairy farmers with varying types of personal protective equipment and handwashing. The intervention “All” refers to farmers wearing gloves, masks, and regularly handwashing. The intervention “No PPE” refers to farmers wearing no masks or gloves and not regularly handwashing.

**Figure 2 microorganisms-11-02413-f002:**
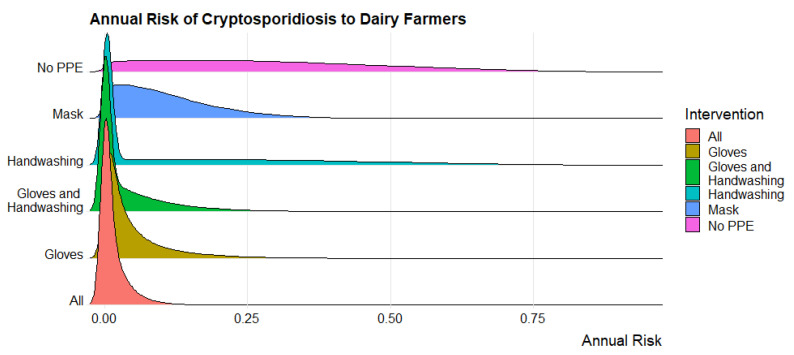
Ridge plot displaying annual risk of cryptosporidiosis to dairy farmers with varying types of personal protective equipment and handwashing. The intervention “All” refers to farmers wearing gloves, masks, and regularly handwashing. The intervention “No PPE” refers to farmers wearing no masks or gloves, and not regularly handwashing.

## Data Availability

Sampling data will not be published as per privacy concerns for the farms. Considering the sample size and small geographical area of the study, researchers are concerned that the farms may be able to be identified by herd size and age.

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
