# Peer review of "Is Personal Protective Equipment Worth the Hassle? Annual Risk of Cryptosporidiosis to Dairy Farmers and How Personal Protective Equipment and Handwashing Can Mitigate It"

_microorganisms, 2023, doi:10.3390/microorganisms11102413_

Round 1

Reviewer 1 Report

The manuscript describes the results of a study on an interesting topic, which is the risk assessment for a specific disease (cryptosporidiosis) and how to promote preventive measures to prevent this risk. However, many methodological aspects are not sufficiently clear, including the research question at the basis of the study. The main points are illustrated here below.

The main objective of the study is not sufficiently clear and what Authors describe at lines 149-151 is simply what they did (the activity), not what they were willing to achieve (the goal). This part should be reformulated, making explicit the research question.

If the main goal was to assess the risk, creating the QMRA model (as I suspect), it’s unclear the usefulness of the copromicroscopic survey (and consequently of the results reported in the paragraph 3.1), since it seems to contribute only with the quantification of cryptosporidium oocysts in the cattle feces. I think that the need for this parameter (or, more generally, for the sampling survey) should be better justified. Moreover, I suspect that other studies (probably wider ones) already investigated prevalence and abundance of Cryptosporidium in cattle in areas similar to the study area or directly in the same area.

Some aspects of the model seem to be not properly addressed.

·      Table 1 is useful in understanding the parameters used, however a formula or a clearer presentation on how they are applied in the model would be helpful.

·      The estimated values of the annual risk (par. 3.2.) do not have any associated uncertainty, which is not realistic. We can guess the uncertainty from the values reported in Table 1 (does “Std” stay for Standard error or for Standard deviation?) and from the two figures, however it’s important to report also the values of these uncertainty intervals, to be scientifically sound. We need to know the value of the estimation, but also how it is precise.

·      Parasites nearly always have an aggregate distribution (negative binomial) and not a “normal distribution”, therefore the assumption taken for the variable “C. parvum in cow feces” is not correct in my opinion.

·      Why oocysts were detected by a molecular method, and not by other methods such as copromicroscopy with staining or by coproantigens, at least in association? The criteria used to decide the diagnostic method should be provided.

·      I don’t see the prevalence among the parameters in the table. Why the prevalence, which is obviously making difference in the risk, is not included in the model?

One on the main innovative aspects of the study is the QMRA model to assess the annual risk for cryptosporidiosis infection in dairy farmers, which is an interesting aspect of this manuscript. However, the model should be scientifically sound and realistic, to be properly used in health education, as proposed by Authors. The finding of the proposed model that the annual risk with no PPE (which seems to be the actual situation in the field) is nearly 100% imply an expected 100% seroprevalence in dairy farmers. To my knowledge, this finding does not match with epidemiological literature data, partly reported by the same Authors at lines 35-37.

I suggest to deeply revise the whole manuscript, revising cautiously the model (starting from the above-mentioned points), reducing strongly the Introduction part, and focusing the Discussion on the relevant point, which is the results of the model, and its possible application. The absence of any structured data collection on the practices in use is also an important shortcoming of the study, although Authors tried to justify it (lines 302-305).

Other minor points should be addressed. Here below some further comments.

Title: I was thinking on a hidden meaning (maybe ‘way of saying’) for “annular risk”, but I finally decided that it was probably a typo, in spite of “annual risk”. Check it.

Introduction: generally, Introduction part is too long and difficult to follow. It goes too much inside life cycle details, it’s redundant in some parts (e.g., lines 67-71 repeat what said in the first paragraph at lines 27-37), and reports unnecessary details of the dairy production system (e.g., lines 84-86). Some parts on the PPE (lines 114-119; 121-128) are either not strictly necessary or can be better placed in the Discussion section.

Line 45: the term “oocyst spores” is new to me. Do Authors mean “sporozoites”?

Lines 71-73: to which nation or state are Authors referring in these lines?

M&M

Line 159: which criteria were used to decide how many individual samples to collect from each involved farm?

Line 160: why fecal samples were not collected directly from the rectum?

Line 163: why stored at -20°C, which is going to affect oocysts? Any reference to justify this?

Line 187: I understand that there is a citation, but it would be beneficial for readers to have a brief description here on how Evers et al. (2014) estimated the amount of feces transferred to an adult’s lips while on farm.

Author Response

Thank you, please find the authors' responses attached. 

Reviewer 2 Report

Overall, nice article that is needed for the industry. I commend the authors for putting this together. Below are some comments for consideration.

One of the major limitations from the model is did you account for the practices performed on each dairy to establish the baseline prevalence? For example, if a farm was already washing hands frequently and wearing gloves, the prevalence of Crypto should already be lower, and thus the results of the model may be underestimating the magnitude of results. Maybe you address this in the first part of the discussion though? 

Introduction is long. Good information, but may review what all is necessary to include for purposes of the current study. May consider reducing.

Line 31: Change "fatality rates" to "case fatality risk." Rate has to involve a time component. Please search and change throughout.

Line 73: reword beginning the sentence with "fully" as is awkward sentence.

Line 160: Defecate instead of defection?

Line 295: Also suggest include how these results may or may not be applicable to much larger dairies in other regions in the country (such as Kansas, California, etc). There are large dairies in those regions that are not corporate. 

Author Response

Thank you, please find our response attached. 
